# Comparative Study of Ozonated Olive Oil and Extra Virgin Olive Oil Effects on Oral Hygiene

Ramona Feier [1], Radu Mircea Sireteanu Cucui [1], Ramona Flavia Ratiu [1], Dana Baciu [1], Carmen Galea [1], Liliana Sachelarie [2,*], Claudia Nistor [1], Dorin Cocos [1], Loredana Liliana Hurjui [3] and Eduard Radu Cernei [3]

1. Department of Medical Disciplines, Faculty of Dental Medicine, University of Targu Mures, 540099 Targu Mures, Romania
2. Department of Preclinical Disciplines, Faculty of Dental Medicine, Apollonia University, 700511 Iasi, Romania
3. Department of Medical Disciplines, Faculty of Dental Medicine, "Grigore T. Popa" University of Medicine and Pharmacy, 700115 Iasi, Romania
* Correspondence: lisachero@yahoo.com

**Abstract:** (1) Background: Currently, more and more studies reveal the fact that the use of ozonated oil in dentistry brings visible benefits. The aim of this study was to evaluate the efficacy of ozonated olive oil by evaluating daily index changes (2) Methods: The available products were used in this study: ozonized olive oil (Ozon Relive) and organic cold-pressed extra virgin olive oil. At the start of the study, all mouthwashes are placed in the same type of containers and labeled with numbers from 1 to 20. The recommended dose is one teaspoon of oil, about 6 mL per day. (3) Results: Ozonated oil (Group 1) and olive oil (Group 2) groups showed statistically significant differences to oral indices ($p < 0.001$ in both). (4) Conclusions: The results suggested that ozonized olive oil can be fully included among the products able to assist in controlling the causative factors of gingivitis while reducing its clinical manifestations.

**Keywords:** oil; ozonized; hygiene; dental

## 1. Introduction

One of the main reasons for the appearance of gingivitis is the accumulation of bacterial plaque [1]. Excess bacterial plaque due to poor dental hygiene determines the apparition of gingival inflammation and leads to periodontal disease. Incorrect habits of dental hygiene determine the appearance of oral diseases, involving both tooth structures and periodontal support [2,3]. Periodontitis is a serious infection of the gums that damages the soft tissue and, without treatment, can cause tooth loosening or lead to tooth loss [4]. The oral cavity is an open and very complex system in which there is a continuous balance between nonpathogen and pathogen microorganisms, depending on the general health status of the subject, and the potential of the intrinsic factors such as immune response to maintain this equilibrium. The disruption of this equilibrium causes the appearance of diseases located at different levels. Bacteria present in dental plaque adhere to hard surfaces or epithelial surfaces with difficulty to be eliminated in absence of specific and regular hygienic measures [4,5]. If the mechanical removal is carried out with the use of various complementary measures such as the use of antibiotics, it was found that it is not enough. Thus, these methods considered conventional have been replaced by brushing, flossing, rinsing, and dental hygiene appointments, all of which are common and mandatory ways to keep bacterial colonies under control [6]. An important aspect is the increase in resistance to antibiotics and therefore an alternative option is being pursued to fight against different microorganisms. The importance of oral health is increasing, being well known that there is an interconnection between oral health and general health, and vice-versa. In the last decade an increasing range of different products either synthetic or natural has invaded the market with the promise of a healthier oral cavity [7]. Such a wide range of products

requires dental operators, dentists, and dental hygienists to give precise indications to patients on the choice and use of these chemical aids [1,2]. The possibility of stopping and reversing the formation of the bacterial plaque and therefore the consequent inflammation without the use of invasive materials seems today to be something more suitable and feasible compared to the possibilities previously entrusted exclusively to the patient's desire to carry out the treatments [8–10]. The occurrence of some pulpal and periapical diseases may be a result of microorganisms eliminated from the dentinal tubules that sometimes might reinfect the root canal. Preventing and stopping microbial development in the oral cavity can be achieved by using antimicrobial solutions such as Chlorhexidine (CHX) a commonly used antiseptic mouthwash used by dentists and the public for its antimicrobial effects [9]. One of these solutions is ozonated olive oil [10,11].

One of the most common inflammatory diseases of the gingival tissue is gingivitis caused by bacterial infection. This is located at the level of soft gingival tissue and conjunctive tissue. The most common variant encountered in the population is the one caused by plaque. Excess bacterial plaque left due to poor dental hygiene implies the appearance of dental plaque. An incorrect practice of dental hygiene involves the appearance of oral diseases [3].

Clinically this presents as swelling, redness, tenderness, a shiny surface, and bleeding on gentle probing. The manifestation is painless so many patients often do not recognize the disease and fail to seek attention [3]. Potentially associated with gingivitis are species of microorganisms such as Streptococcus, Fusobacterium, Actinomyces, Veillonella, Treponema, Bacteroides, Capnocytophaga, and Eikenella. Of course, other etiological factors favor faster plaque deposition or the tissue's vulnerability to microbial attack [3,4]. Some of the types of bacteria found within oral infections are part of the microbiota of the oral cavity. As the periodontal disease gingivitis is more common in men than in women, because women have better oral care [12].

In the treatment of oral infections, the correct choice of antibacterial products has a double benefit, namely the rapid eradication of the infection, which decreases the degree of tissue destruction; as well as decreasing the use of inappropriate antibiotics to prevent the development of antibiotic resistance. Antibiotics containing the narrowest spectrum should be selected based on the results of culture and susceptibility testing [5,6].

As far as the oral cavity is concerned, it is an open complex in which there is a balance between the entering microorganisms and the ways of defending the host to eliminate them. Bacteria adhere to hard surfaces or to epithelial surfaces to try not to be eliminated [4,5]. If their mechanical removal is carried out with the use of various complementary techniques such as antibiotics, it was found that it is not enough [7]. Antibiotic therapy is a powerful adjunct to conventional mechanical debridement for the therapeutic management of the periodontal disease [9]. Mechanical treatment and combined surgical treatment can halt or prevent further periodontal attachment loss in most individuals by reducing the total supra-subgingival bacterial mass [4].

Mechanical periodontal treatment supports the host's defense system in overcoming the infection by killing subgingival pathogens, and systemic antibiotics can significantly enhance the effects of mechanical periodontal therapy in combination with measures that improve oral hygiene [9].

Olive oil is a vegetable oil obtained by pressing ripe olives. Ozoneted olive oil acts differently from traditional drug-receptor interactions but acts as a generator of effector molecules such as various small hydrophilic and lipophilic molecules with selective action. Pathogenic bacteria from an electrical point of view create a positive charge and the ozone that carries an electrical charge is attracted to it and therefore tries to eliminate the infection and disinfect [5–7].

Ozonated olive oil has changed in its chemical composition and new biological properties. Most of the fatty acids in olive oil, most of them, are monounsaturated, predominantly oleic acid (65–85%). With the exception of fatty acids, olive oil is a source of bioactive compounds such as tocopherols or phenolic compounds, with the exception of fatty acids [2].

In the ozonation process, ozone interacts mainly with carbon-carbon double bonds in unsaturated fatty acids giving peroxides, aldehydes and ozonides, which are considered to be responsible for an improvement in the biological activity of the ozonated oil. The amounts and ratio of the compounds depend on several factors such as ozonation time, process temperature and reactor type. According to the literature data, ozonation increases the peroxide value with the amount of ozone introduced to the matrix [11].

The introduction of ozone in dental practice is a new alternative. It was found that thanks to oxygen, has a beneficial role in the treatment of oral diseases [12].

The possibility of stopping and reversing the formation of the bacterial plaque and therefore the consequent inflammation without the use of invasive materials seems today to be something more concrete and feasible compared to the possibilities previously entrusted exclusively to the patient's desire to carry out the treatments provided by the previous operating protocols and to his genetic characteristics [13,14]. The occurrence of pulpal and periapical diseases is the result of microorganisms that remain in the dentinal tubules being eliminated and which reinfect the root canal. Preventing and stopping microbial development can be achieved by using antimicrobial solutions. One of these solutions is ozonated olive oil [11,14]. The use of ozone in the dental field has become more frequent in recent years due to its high oxidative power that induces a high immune response and blood circulation, as well as a strong antimicrobial activity [15].

Ozonated oil can be used in practice and the home dental routine, providing an added advantage for high-quality oral treatment at home. Practically, any process caused by bacteria in the mouth can be solved by using ozonated olive oil [16,17]. It applies ozonated olive oil in periodontal therapy, in root canals to kill bacteria in root canal-treated teeth. When doing a crown or filling, we use ozone before placing the crown or filling and it is very useful when we do extractions to kill all the bacteria and viruses in a socket before we finish. Any bacterial infection can be effectively treated with ozonated olive oil [18–21]. Studies have shown a decontaminant effect of ozone applied on the tooth prior to placing either a crown or filling and in the socket site post extraction. Any bacterial infection can be effectively treated with ozonated olive oil [16].

The present study aims to demonstrate the efficacy of a new product based on ozonated olive oil in the control of bacterial plaque and the inflammatory response in gingivitis [1].

## 2. Materials and Methods

### 2.1. Aim of the Study

The aim of this study is to evaluate the efficacy of ozonated olive oil in dental hygiene by comparing it with pure olive oil.

### 2.2. Materials

The study was conducted in Targu Mures, in the Department of Dentistry Faculty, Dimitrie Cantemir University of Targu Mures, Romania, in according to The Declaration of Helsinki and approved by the Ethics Committee of Dimitrie Cantemir University (51/18.03.2022).

The study included 71 patients with gum disease. Subjects were divided according to therapy. After the professional hygiene treatment, patients are divided into 2 groups according to the product they will have to use for home hygiene: group 1–31 patients and ozonated olive oil and group 2–30 patients and extra virgin olive oil.

### 2.3. Inclusion Criteria

We included patients aged between 27and 54 years with the presence of:

- minimum 20 teeth, excluding third molars
- PPD $\leq$ 3 mm
- absence of periodontitis and orthodontic appliances, without known allergy to any of the mouthwash components

*2.4. Exclusion Criteria*

Subjects with the following characteristics were excluded from the study:

- last periodontal treatment < 6 months
- patients suffering from systemic pathologies that may influence therapy such as: diabetes, neoplasms, pathologies of bone metabolism, disorders that compromise healing, radiation, immunosuppressive therapies, and anticoagulant therapies
- patients affected by periodontal disease PPD (periodontal probing depth) > 3 mm
- taking antibiotics < 6 months; anti-inflammatory drugs < 3 months; cortisone; contraceptive hormones

Smoking is reported but is not an exclusion criterion.

*2.5. Methods*

The study is a pilot, randomized, double-blind clinical trial. It is conducted in accordance with the ethical standards set forth in the Declaration of Helsinki and informed consent was obtained from all participants in the study. The samples were opaque, sealed, and opened just before the treatment. In order to eliminate the possible subjective evaluation error, the study is conducted in a double-blind manner. The available products were used in this study: ozonized olive oil (Ozonrelieve, Rome, Italy) and organic cold-pressed extra virgin olive oil. At the start of the study, all mouthwashes are placed in the same type of containers and labeled with numbers from 1 to 20. The recommended dose is one teaspoon of oil, about 6 mL per day.

In each patient, we analyzed the daily changes of the index, the Full Mouth Plaque Score (FMPS—the presence of plaque is scored as 1, the absence as 0; the average plaque value is shown as a percentage—% of the sites detected as positive for the presence of plaque. out of a total of 6 sites per tooth), the Full Mouth Bleeding Score (FMBS—the average value of bleeding is indicated as a percentage of the bleeding sites from the total number of sites examined—6 per tooth) and the gingival index (GI—a value of to 0 to 3 is assigned to the four sites of each element mesial, distal, buccal and lingual/palatal).

The oral indices were analyzed at the first consultation, marked V0, and after the treatment with ozonated olive oil, three weeks apart, at a new consultation V1.

All patients had significant plaque accumulation as evidenced by the FMPS, despite reporting regular use of a toothbrush 3 times a day and flossing or interdental brushing once a day for interproximal hygiene. FMBS values, the index of inflammation in the active phase, are also above the threshold value, of 20%. The gingival index, GI, makes it possible to evaluate the degree of gingival inflammation in relation to the morphological parameters of changes in color, shape, and consistency. The subjects examined presented a homogeneous clinical picture, the values actually describing mild inflammation in almost all subjects.

*2.6. Statistical Analysis*

The statistical analysis was performed using IBM SPSS statistics software version 20. Using *t* Test, descriptive statistics were performed before and after using ozonized olive oil. Data are presented as mean $\pm$ standard deviation (SD) or percentage for categorical variables and *p*-value for statistical significance.

## 3. Results

*3.1. Characteristics of the Population (Age, Gender, and Environment)*

In our study group over 50% were women, the average age of the group 1 (age between 27–53 years old) was 40.33 $\pm$ 9.09 years, in group 2 (age between 27–54 years old) was 40.62 $\pm$ 0.16 years, and the patients came mainly from the urban environment (54.92%).

In our study group over 50% were women, the average age of the group 1 (age between 27–53 years old) was 40.33 $\pm$ 9.09 years, in group 2 (age between 27–54 years old) was

40.62 ± 0.16 years, and the patients came mainly from the urban environment (54.92%), (Table 1).

**Table 1.** Characteristics of the population.

| Baseline Characteristics of the Two Groups | Group 1 Ozonated Olive Oil MD ± DS | | Group 2 Extra Virgin Olive Oil MD ± DS | |
|---|---|---|---|---|
| Age (years) | 40.33 ± 9.09 | | 40.62 ± 0.16 | |
| Gender | | Percentage% | | Percentage% |
| Women | 21 | 51.21 | 20 | 48.78 |
| Men | 17 | 56.66 | 13 | 43.34 |
| Environment | | | | |
| Urban | 27 | 38.02 | 12 | 61.98 |
| Rural | 22 | 30.98 | 10 | 69.01 |

### 3.2. Indices Changes

All patients present significant plaque accumulation, as evidenced by FMPS, at the first visit Vo. FMBS values, the index of inflammation in the active phase, are also above the threshold value of 20%. The gingival index, GI, makes it possible to evaluate the degree of gingival inflammation in relation to the morphological parameters of changes in color, shape, and consistency. The subjects examined presented a homogeneous clinical picture, the values actually describing mild inflammation in almost all subjects (Figure 1).

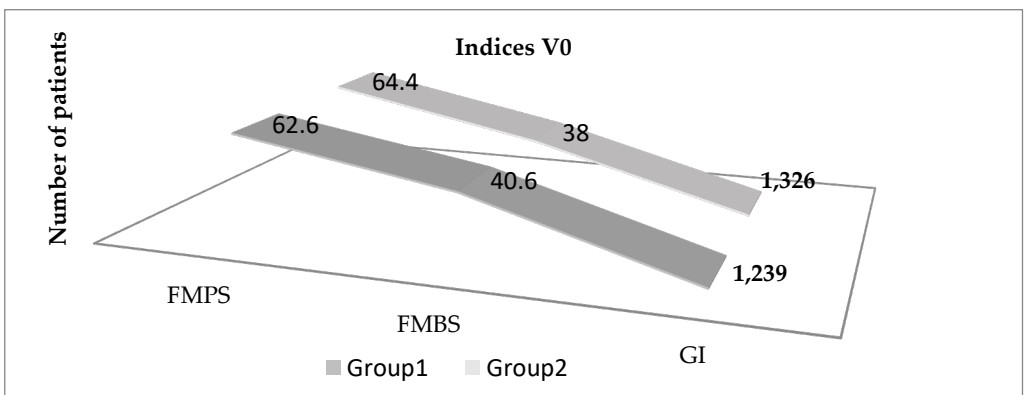

**Figure 1.** The average value of oral indices at V0 for the two groups.

After the occupational hygiene session, patients were asked to maintain the same occupational hygiene methods, in order not to determine an additional variable, integrating them with the use of the allocated mouthwash, for 3 weeks.

### 3.3. Variation of Indices from V0 to V1

The new data collected at the next visit V1, after 3 weeks show an improvement in the three indices in both groups but the most in group 1 Figure 2.

In the case of patients who used the ozonated oil, the mean value of the GI index decreased from 1.291 (at V0) to 0.876 (V1) with an absolute improvement of 0.415. In the case of patients who used pure oil, the average value of the GI index decreased slightly from 1.326 (at V0) to 1.093 (V1) with an absolute of 0.233.

In the case of patients who used the ozonated oil, the mean value of the FMPS index decreased significantly from 62.6 (at V0) to 20.9 (V1) with an absolute improvement of 41.7. In the case of patients who used pure oil, the average value of the FMPS index decreased from 64.4 (at V0) to 31.4 (V1) with an improvement of 33.

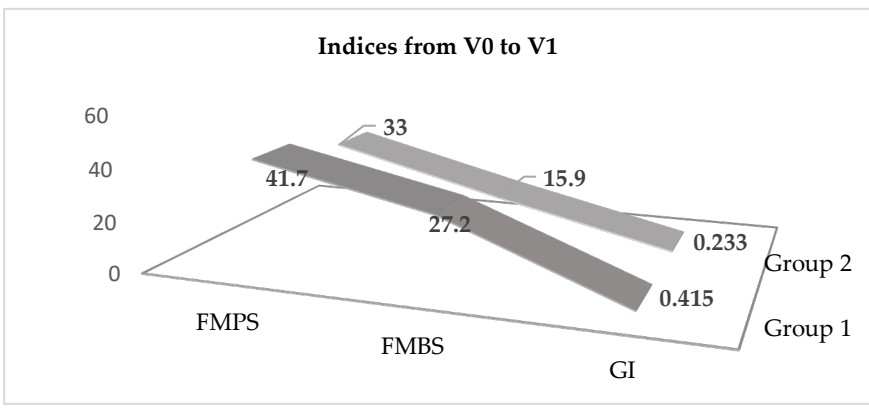

**Figure 2.** Variation of oral indices from V0 to V1.

In the case of patients who used the ozonated oil, the mean value of the FMBS index decreased significantly from 40.6 (at V0) to 13.4 (V1) with an absolute improvement of 27.2. In the case of patients who used the pure oil, the average value of the FMBS index decreased from 38 (at V0) to 22 (V1) with an improvement of 15.9.

The new data collected at V1 were compared with those from V0 and an improvement in the three indices considered can be seen in both groups.

To evidence the effect of the therapy, the differences of the variables at time V1 and at time V0 are taken into account (Tables 2–4).

**Table 2.** Descriptive statistics for Group 1.

| | | | | | Descriptives Statistics *t*-Test | | | | |
|---|---|---|---|---|---|---|---|---|---|
| | | | | | 95% Confidence Interval for Mean | | | | |
| **Ozonated Olive Oil Group 1** | **N** | **Mean** | **Std. Deviation** | **SEM** | **Lower Bound** | **Upper Bound** | **T** | ***p* Value** | **Df** |
| FMPS 0 | 31 | 112.10700 | 150.06167 | 33.55481 | 22.90983 | 159.50017 | 2.7035 | 0.0102 | 58 |
| FMPS 1 | 31 | 20.90200 | 15.62014 | 3.49277 | | | | | |
| FMBS 0 | 31 | 40.59100 | 17.84195 | 3.25748 | 20.12495 | 34.28505 | 7.6916 | 0.0001 | 58 |
| FMBS 1 | 31 | 13.38600 | 7.54811 | 1.37809 | | | | | |
| GI 0 | 31 | 1.3094 | 0.3028 | 0.0535 | 0.3119 | 0.6040 | 6.2758 | 0.0001 | 58 |
| GI 1 | 31 | 0.8514 | 0.2560 | 0.0484 | | | | | |

Statistical analysis of oral index values of group 1 from the first consultation (Vo) to the second consultation after three weeks (V1) indicates a very statistically significant ($p < 0.001$).

**Table 3.** Descriptive statistics for group 2.

| | | | | | Descriptives Statistics *t*-Test | | | | |
|---|---|---|---|---|---|---|---|---|---|
| | | | | | 95% Confidence Interval for Mean | | | | |
| **Olive Oil Group 2** | **N** | **Mean** | **Std. Deviation** | **SEM** | **Lower Bound** | **Upper Bound** | **T** | ***p* Value** | **Df** |
| FMPS0 | 30 | 150.96484 | 282.59950 | 50.75637 | 19.96657 | 119.9935 | 2.3988 | 0.0195 | 61 |
| FMPS1 | 30 | 30.97125 | 15.65427 | 2.76731 | | | | | |
| FMBS 0 | 30 | 37.24516 | 20.18458 | 3.62526 | 6.27238 | 24.69601 | 3.3623 | 0.0014 | 60 |
| FMBS 1 | 30 | 21.76097 | 15.81244 | 2.84000 | | | | | |
| GI 0 | 30 | 1.3260 | 0.2796 | 0.0510 | 0.1002 | 0.3658 | 3.5128 | 0.0009 | 58 |
| GI 1 | 30 | 1.0930 | 0.2320 | 0.0424 | | | | | |

**Table 4.** Descriptive comparing statistics between Group 1 and Group 2.

| Comparing Scores between Group 1 and Group 2 V0-V1 | T | Df | Sig. | Mean Difference | 95% Confidence Interval for Mean | |
|---|---|---|---|---|---|---|
| | | | | | **Lower Bound** | **Upper Bound** |
| FMPS | 0.5625 | 59 | 0.005 | −0.01800 | −0.08206 | 0.04606 |
| FMBS | 7.4371 | 59 | 0.0001 | 0.27200 | 0.19882 | 0.34518 |
| GI | 3.2860 | 59 | 0.0017 | −0.21700 | −0.34914 | −0.08486 |

Comparing Scores between Group 1 and Group 2 indicates a statistically significant ($p < 0.001$) and a beneficial action of the ozonated olive oil.

## 4. Discussion

The oral cavity can be represented as an open ecosystem, able to guarantee a constant dynamic balance between the penetration of bacterial, viral, or fungal microorganisms and their colonization methods, nutritional balance, and established defenses against removal. The antimicrobial effects and anti-inflammatory properties of ozonated olive oil have been demonstrated in several studies [11,12].

In dental practice, there is a multitude of treatment options available for all oral lesions and conditions. The use of ozonated olive oil is a minimally invasive technique that can be used for these conditions without side effects [5].

Our study shows that the use of ozonated olive oil in oral hygiene has beneficial effects. By decreasing the values of oral indices in patients who used ozonated olive oil, it is confirmed that it is one of the good products to use in oral hygiene.

Antimicrobial activity varies from one microorganism to another and from one essential oil to another, but is always dose-dependent [3,4,11]. It is strictly related to the chemical composition and concentration of their constituents, which do not depend only on the species, but also on other factors such as the origin of the plant, the part used, the stage of development, growth (temperature, soil, fertilizers, etc.), distillation and storage conditions [12–15].

The antimicrobial activity, reflected by the index value, depends on the chemical composition. The mechanism of action of essential oils against microorganisms is complex and has not yet been well clarified because it depends on various factors: the type of antimicrobial potency of different essential oils which in turn depends on their chemical composition and therefore on their predominantly hydrophilic or lipophilic characteristics; according to the type of microorganisms and is mainly related to the structure of their cell wall [16–18]. Due to the variability of the amounts and components of essential oils, it is very likely that their antimicrobial activity is not due to a single mechanism, but to different modes of action at the cellular level [19]. A series of research shows that ozonated oil is an antiseptic and Gram-negative bacteria have proven to be more sensitive to ozonated olive oil than Gram-positive ones [20,21] Nagayoshi et al demonstrated that ozonated water strongly inhibits plaque accumulation in vitro, with high efficacy against Gram-positive and negative oral microorganisms [22–24]. Also, the potential antimicrobial activity has been investigated against Streptococcus mutans, Lactobacillus rhamnosus and Candida albicans with great effectiveness regarding the use of ozonated olive oil [22].

An important characteristic of essential oil components is actually their hydrophobicity, which allows essential oils to partition between bacterial or fungal cell membrane lipids and mitochondria, altering cell structures and thus making them more permeable. Ozone, in the early oxidizing process, attacks glycoproteins and other amino acids, and inhibits the enzyme control system of the main cells by creating permeability in the membranes due to ozone [13]. Excessive losses of ions and molecules from the microbial cell will inevitably lead to death [23–25]. Essential oils appear to act predominantly with structural and functional changes in fungal membranes, leading to cytoplasmic dispersion and cell death. There is therefore a blockage of membrane synthesis, inhibition of germination, reproduction, and cellular respiration [26].

The ozonation reaction applied to olive oil arises from the need to "deposit" ozone in a biological substrate compatible with biological tissues capable of maintaining unchanged the therapeutic properties of the molecule and its intermediates [27]. The ozonated oil used in the topical modality represents an absolutely non-invasive intervention technique that can also be used for various clinical conditions without presenting unwanted side effects. Taking into account what has been described ozonated oil can represent and guarantee a determining factor in the treatment of gingivitis and oral lesions [22]. Studies have shown that after treatment, plaque index and bleeding index differ significantly from the indices of placebo-treated subjects in any comparison [28].

The ozonated oil may play a key role in the treatment of gingivitis and oral lesions, and there is also some preliminary evidence to suggest that it reduces the levels of several

compounds associated with halitosis [26]. The studies carried out in the field of dentistry have led to considerable progress in the field of prevention and hygiene, with positive results, and therefore the use of ozonated oil guarantees this turning point in dental treatments [25,26]. The study showed that ozonated olive oil, due to its organoleptic properties, the absence of toxicity and side effects, and its biocompatibility, represents a choice in the treatment of various oral diseases. Ozone has the power to kill bacteria, viruses, and fungi as well as other pathologies [27,28]. Studies show that it provides soothing treatment for fungal infections, dermatitis, infection sites, periodontal pockets, and more. Nagayoshi et al demonstrated that ozonated water strongly inhibits plaque accumulation in vitro, with high efficacy against Gram-positive and negative oral microorganisms [29]. Ozonated olive oil offers a quick and easy way to apply ozone directly where it is needed and facilitates healing. People using ozonated olive oil for teeth have noted success against tooth decay, gingivitis, and chapped lips [30,31].

The release of ozone can be achieved in several ways because it becomes unstable when dissolved in water, it decomposes quickly through a complex series of chain reactions, so it cannot be stored [32]. Olive oil reacts with ozone thus forming long complex molecules [27]. Several studies confirm the fact that the use of ozonated olive oil gel has a longer duration of action in the oral cavity with beneficial effects in the treatment of oral diseases [28].

Ozone appears to strongly inhibit the formation of dental plaque and reduce the number of pathogens [16]. The oxidative power of ozone is 1.5 times that of chloride when used as an antimicrobial agent [6]. This oxidation effect gives ozone its bactericidal, virucidal, and fungicidal activity [12]. According to microbiological studies, ozone is able to kill Gram-positive and Gram-negative bacteria, including *Pseudomonas aeruginosa* and *Escherichia coli* [6]. This antimicrobial capacity is the result of the effects of ozone on cells such as damage to the cytoplasmic membrane due to ozonolysis of double bonds and induction of cytoplasmic change contents. This action does not seem to harm the human body cells; the reason attributed to this is the antioxidant capacity of mammalian cells [18].

The therapeutic action of ozonated olive oil is its effectiveness in reducing periodontal indices. Subjects using ozonated olive oil for teeth have noted good responses in diminishing the rate of tooth decay, gingivitis, and chapped lips and ozonated olive oil could be considered an alternative antibacterial agent [28,29].

There are also studies that have shown that ozonated olive oil promotes the healing of grafts with a very good prognosis [30]. A particularly important aspect of dentistry is the prevention of various diseases. There are studies that indicate that ozonated olive oil has the effect of improving the results of scaling and root planing (SRP) in the treatment of periodontal diseases [31,32]. Compliance with tooth brushing procedures and the use of mouthwash with ozonated olive oil could be an additional preventive tools for the oral hygiene procedure at home. Further research is needed to evaluate the effectiveness of ozonated olive oil mouthwashes in patients at high risk of caries and subjects requiring orthodontic therapy [33–35].

One of the study's limitations is the small number of groups of study. Further research is needed to evaluate the effectiveness of ozonated olive oil mouthwashes in patients at high risk of caries and subjects requiring orthodontic therapy [36,37].

Through acceleration of healing, reduction of microbial infection, modulation of the inflammation phase, and enzymatic reactions in oxygen metabolism ozonated olive oil represents an efficient and at the same time cheap alternative method that must be used in the dental field [38–40].

The results obtained are given by the different components of olive oil, responsible for the bactericidal action and for skin absorption because it can stabilize ozone, which is a very reactive molecule [34]. Ozonated olive oil combines the properties of ozone with those of olive oil to achieve an unparalleled compound. The oxidizing power of ozone has interesting effects on microorganisms and wound healing [41–43].

There is a lack of literature regarding the effect of ozone oil versus olive oil on the clinical parameter of periodontal inflammation. Due to its beneficial biological properties,

including antimicrobial and immune-stimulating effects, the use of ozonated olive oil in dentistry has opened up new perspectives in the treatment of dental pathologies for patients of all ages.

## 5. Conclusions

The formulation of the tested product, according to the indications of the manufacturer, would allow it to obtain an antimicrobial and anti-inflammatory effect. The data obtained, despite the small sample quantities, indicate efficacy in the treatment of gingivitis. Our clinical investigations confirmed the effectiveness of mouthwash is expressed both by an antibacterial action and by reducing the inflammatory response and bleeding. The proposed work shows that ozonated olive oil can be fully included among the products able to assist in controlling the causative factors of gingivitis while reducing its clinical manifestations. All these values confirm the therapeutic possibilities of the tested product.

**Author Contributions:** Conceptualization, R.F., R.F.R. and L.S.; methodology, R.M.S.C.; software, L.S.; validation, D.B., C.G., and E.R.C.; formal analysis, D.C.; investigation, D.B.; resources, R.M.S.C.; data curation. C.N., C.G. and D.C.; writing—original draft preparation, L.S., R.M.S.C.; writing—review and editing, L.S., L.L.H.; visualization, C.N., C.G. and D.C.; supervision, L.S.; project administration L.S., R.F., R.F.R. All authors have read and agreed to the published version of the manuscript.

**Funding:** This research received no external funding.

**Institutional Review Board Statement:** The study was conducted according to the guidelines of the Declaration of Helsinki, and approved by the Ethics Committee of Dimitrie Cantemir University (51/18.03.2022).

**Informed Consent Statement:** Not applicable.

**Conflicts of Interest:** The authors declare no conflict of interest.

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
