# Peer review of "Comparative Study of Ozonated Olive Oil and Extra Virgin Olive Oil Effects on Oral Hygiene"

_applsci, doi:10.3390/app13052831_

Round 1

Reviewer 1 Report

The paper entitled “Antibacterial and anti-inflammatory effects of ozonated olive oil in dentistry” is an original article reporting the effects of olive oil versus ozonized oil to control dental plaque in a sample of 71 subjects with “gum diseases” not well classified and specified.

The manuscript requires revision for the language, which is not always correct in grammar and syntax. 

The introduction is chaotic and not orderly developed.

I suggest reorganizing its content by following the following points or similar:

  1. The bacterial plaque: pathogenic features
  2. Standard protocols to control plaque growth and maturation (prevention) and remotion (treatments)
  3. The limitations of the common substances (antibiotics, chlorhexidine, and other antiseptics)
  4. The evidence from the literature about ozone, olive oils, and ozonated oil and the differences among them
  5. Focus on ozonated oils (properties, beneficial effects..)
  6. The literature report (state of the art) on ozonated oil in dentistry
  7. Aims of the study 

 Lines 37-38 “An important current aspect is the increase in resistance to antibiotics and therefore an alternative option is being pursued to fight against different microorganisms.”. 

Before defining the novel treatments, the authors should discuss the state of the art on oral antiseptics available and used. Please, focus mainly on chlorhexidine and antibiotic treatments. Only in this context, the authors can introduce the limitations of these drugs (antibiotic resistance and changes in the oral microbiota composition), justifying the need for new products, such as ozonized olive oil. 

Lines 59-61 “Ozone acts differently from traditional drug-receptor interactions but acts as a generator of effector molecules such as various small hydrophilic and lipophilic molecules with selective action.”

“various small molecules”..be more detailed. 

The authors should add a figure schematizing the mechanisms of action of ozone and ozonized oils or, if unknown by the literature, the composition of the compounds. 

Line 67” as a prophylactic method and in treatment”. For what kind of condition? Please, be less vague. 

M&M

“gum disease”: too vague. Were the selected patients affected by gingivitis? Periodontitis? Please, add measures of the diseases by periodontal indices. Although the author have reported them later, the paragraph needs to be better rewritten. 

The patients were divided into 2 groups. A third group as the positive control (treated with conventional antiseptics, such as chlorhexidine) and/or a fourth group as negative control (untreated at all) would increase the quality of the study. 

Lines 193-196 are repetitions of the previous three lines. 

Results

“The clinical data collected during the first visit (V0) of the subjects indicated that all 198 patients have significant plaque accumulation as evidenced by FMPS, despite reporting 199 regular use of toothbrush, 3 times a day, and flossing or interdental brush, for inter-200 proximal hygiene, 1 time a day.”.

It was an inclusion criterium reported in M&M; it is not a result. 

Figure 1: Gro… what does it mean? “group 1 and 2”? Please, mind the figure and the content. 

Line 214 “”after 3 weeks” of what? Treatment? Or 3 weeks for dental hygiene?

the protocol was not clear :

At V0, the periodontal indices were registred. 

Then a full mouth professional hygiene? How was it eventually performed?

When did the administration of oil/ozonized oil start?

What are the differences between ozonized and hyper-ozonized oils?

The results and their statistical significance require a better description. 

In table, p 0,000 is unacceptable. 

Discussion: it needs a full revision: it is redundant, vague, and inconclusive in the first half. The second half should better fit in the introduction to explain the state of the art of ozonized oils used in dentistry. Otherwise, the authors could reorganize both the introduction and the discussion in a more flow and transparent way. 

Author Response

The authors acknowledge the useful observations and suggestions of the reviewer’s as concerns the manuscript entitled:  Comparative study of ozonated olive oil effects on oral hygiene indices

Ramona Feier1, Mircea Radu Sireteanu Cucui1, Ramona Flavia Ratiu1, Dana Baciu1, Carmen Galea1, Liliana Sachelarie 2,*, Claudia Nistor1, Dorin Cocos1, Loredana Liliana Hurjui3, Cernei Eduard Radu3

According to the reviewer’s recommendations, all the suggestions were taken into account, as follows:

The paper entitled “Antibacterial and anti-inflammatory effects of ozonated olive oil in dentistry” is an original article reporting the effects of olive oil versus ozonized oil to control dental plaque in a sample of 71 subjects with “gum diseases” not well classified and specified.

The manuscript requires revision for the language, which is not always correct in grammar and syntax. 

The introduction is chaotic and not orderly developed.

I suggest reorganizing its content by following the following points or similar:

  1. The bacterial plaque: pathogenic features
  2. Standard protocols to control plaque growth and maturation (prevention) and remotion (treatments)
  3. The limitations of the common substances (antibiotics, chlorhexidine, and other antiseptics)
  4. The evidence from the literature about ozone, olive oils, and ozonated oil and the differences among them
  5. Focus on ozonated oils (properties, beneficial effects..)
  6. The literature report (state of the art) on ozonated oil in dentistry
  7. Aims of the study 

Done

 Lines 37-38 “An important current aspect is the increase in resistance to antibiotics and therefore an alternative option is being pursued to fight against different microorganisms.”. 

Before defining the novel treatments, the authors should discuss the state of the art on oral antiseptics available and used. Please, focus mainly on chlorhexidine and antibiotic treatments. Only in this context, the authors can introduce the limitations of these drugs (antibiotic resistance and changes in the oral microbiota composition), justifying the need for new products, such as ozonized olive oil. 

Lines 59-61 “Ozone acts differently from traditional drug-receptor interactions but acts as a generator of effector molecules such as various small hydrophilic and lipophilic molecules with selective action.”

“various small molecules”..be more detailed. 

The authors should add a figure schematizing the mechanisms of action of ozone and ozonized oils or, if unknown by the literature, the composition of the compounds. 

Line 67” as a prophylactic method and in treatment”. For what kind of condition? Please, be less vague. 

M&M

“gum disease”: too vague. Were the selected patients affected by gingivitis? Periodontitis? Please, add measures of the diseases by periodontal indices. Although the author have reported them later, the paragraph needs to be better rewritten. 

Done

One of the most common inflammatory diseases of the gingival tissue is gingivitis caused by bacterial infection. This is located at the level of soft gingival tissue and conjunctive tissue. The most common variant encountered in the population is the one caused by plaque.

Clinically this presents as swelling, redness, tenderness, a shiny surface and bleeding on gentle probing. The manifestation is painless so that many patients often do not recognize the disease and fail to seek attention [3]. Potentially associated with gingivitis are species of microorganisms such as Streptococcus, Fusobacterium, Actinomyces, Veillonella, Treponema, Bacteroides, Capnocytophaga, and Eikenella. Of course, there are other etiological factors that favor faster plaque deposition or the tissue's vulnerability to microbial attack [3,4].Some of the types of bacteria found within oral infections are part of the microbiota of the oral cavity. As the periodontal disease gingivitis is more common in men than in women, because women have better oral care [3].

      One of the main reasons for the appearance of gingivitis is the accumulation of bacterial plaque. Excess bacterial plaque left due to poor dental hygiene implies the appearance of dental plaque. An incorrect practice of dental hygiene involves the appearance of oral diseases [3]. In the treatment of oral infections, the correct choice of antibacterial products has a double benefit, namely the rapid eradication of the infection, which decreases the degree of tissue destruction; as well as decreasing the use of inappropriate antibiotics to prevent the development of antibiotic resistance. Antibiotics containing the narrowest spectrum should be selected based on the results of culture and susceptibility testing [5,6].

      One of the main reasons for the appearance of gingivitis is the accumulation of bacterial plaque. Excess bacterial plaque due to poor dental hygiene determine apparition of gingival inflammation and lead to periodontal disease. Incorrect habits of dental hygiene determine the appearance of oral diseases, involving both tooth structures and periodontal support [3].

      Oral cavity is an open and very complex system in which there is a continuous   balance between non pathogen and pathogen microorganisms, depending by the general health status of the subject, and the potential of the intrinsic factors such as immune response to maintain this equilibrium. The disruption of this equilibrium causes the appearance of disease located at different levels.  Bacteria present in the dental plaque adhere to hard surfaces or to epithelial surfaces with difficulty to be eliminated in absence of specific and regular hygienic measures [4,5]. If the mechanical removal is carried out with the use of various complementary measures such as use of antibiotics, it was found that it is not enough. Thus, these methods considered conventional have been replaced by brushing, flossing, rinsing, and dental hygiene appointments, all of which are common and mandatory ways to keep bacterial colonies under control [6]. An important aspect is the increase in resistance to antibiotics and therefore an alternative option is being pursued to fight against different microorganisms. The importance of oral health is increasing, being well known that there is an interconnection between oral health and general health, and vice-versa. In the last decade an increasing range of different products either synthetic or natural have invaded the market with the promise of a healthier oral cavity [7]. Such a wide range of products requires dental operators, dentists, and dental hygienists to give precise indications to patients on the choice and use of these chemical aids [1,2]. The possibility of stopping and reversing the formation of the bacterial plaque and therefore the consequent inflammation without the use of invasive materials seems today to be something more suitable and feasible compared to the possibilities previously entrusted exclusively to the patient's desire to carry out the treatments [8-10].  The occurrence of some pulpal and periapical diseases may be a result of microorganisms eliminated from the dentinal tubules being that sometimes might reinfect the root canal. Preventing and stopping microbial development in the oral cavity can be achieved by using antimicrobial solutions. One of these solutions is ozonated olive oil [10,11].

Antibiotic therapy is a powerful adjunct to conventional mechanical debridement for the therapeutic management of periodontal disease [9]. Mechanical treatment and combined surgical treatment can halt or prevent further periodontal attachment loss in most individuals by reducing the total supra-subgingival bacterial mass [4].

      Mechanical periodontal treatment supports the host's defense system in overcoming the infection by killing subgingival pathogens, and systemic antibiotics can significantly enhance the effects of mechanical periodontal therapy in combination with measures that improve oral hygiene [9].

Studies shown decontaminant effect of ozone applied on tooth prior placing either a crown or filling and in the socket site post extraction. Any bacterial infection can be effectively treated with ozonated olive oil [16].

The present study aims to demonstrate the efficacy of a new product based on ozonated olive oil in the control of bacterial plaque and the inflammatory response in gingivitis [1].

The patients were divided into 2 groups. A third group as the positive control (treated with conventional antiseptics, such as chlorhexidine) and/or a fourth group as negative control (untreated at all) would increase the quality of the study. 

 It is a very good idea but maybe in another study. Thank you very much!

Lines 193-196 are repetitions of the previous three lines. 

Results

“The clinical data collected during the first visit (V0) of the subjects indicated that all 198 patients have significant plaque accumulation as evidenced by FMPS, despite reporting 199 regular use of toothbrush, 3 times a day, and flossing or interdental brush, for inter-200 proximal hygiene, 1 time a day.”.

It was an inclusion criterium reported in M&M; it is not a result. 

All patients present significant plaque accumulation, as evidenced by FMPS, at the first visit Vo.

Figure 1: Gro… what does it mean? “group 1 and 2”? Please, mind the figure and the content.

Done 

Line 214 “”after 3 weeks” of what? Treatment? Or 3 weeks for dental hygiene?

  3 weeks for dental hygiene

the protocol was not clear :

At V0, the periodontal indices were registred. 

Yes.

Then a full mouth professional hygiene? How was it eventually performed?

When did the administration of oil/ozonized oil start?

From the first visit!

What are the differences between ozonized and hyper-ozonized oils?

We did not use hyper-ozonized.

The results and their statistical significance require a better description. 

In table, p 0,000 is unacceptable. 

Sorry, we remade the statistics!

            To evidence the effect of the therapy, the differences of the variables at time V1 and at time V0 are taken into account ( Table 2, Table 3, Table 4).

                    Table 2. Descriptive statistics for group 1

Descriptives  statistics t-test

Ozonated olive oil     Group 1

N

Mean

Std. Deviation

SEM

95% Confidence Interval for Mean

Lower Bound

Upper Bound

 t

P value

df

FMPS 0

31

112.10700

150.06167

33.55481

22.90983

159.50017

2.7035

0.0102

58

FMPS 1

31

20.90200

15.62014

3.49277

FMBS 0

31

40.59100

17.84195

3.25748

20.12495

34.28505

7.6916

0.0001

58

FMBS 1

31

13.38600

7.54811

1.37809

GI 0

31

1.3094

0.3028

0.0535

0.3119

0.6040

6.2758

0.0001

58

GI 1

31

0.8514

0.2560

0.0484

                                                            Statistical analysis of oral index values of group 1 from the first consultation (Vo) to the                                                        second consultation after three weeks (V1) indicates a very statistically significant                                                                         (p<0.001).

Table 3. Descriptive statistics for group 2

Descriptives  statistics t-test

Olive oil    

Group 2

N

Mean

Std. Deviation

SEM

95% Confidence Interval for Mean

Lower Bound

Upper Bound

 t

P value

df

FMPS0

30

150.96484

282.59950

50.75637

19.96657

119.9935

2.3988

0.0195

61

FMPS1

30

30.97125

15.65427

2.76731

FMBS 0

30

37.24516

20.18458

3.62526

6.27238

24.69601

3.3623

0.0014

60

FMBS 1

30

21.76097

15.81244

2.84000

GI 0

30

1.3260

0.2796

0.0510

0.1002

0.3658

3.5128

0.0009

58

GI 1

30

1.0930

0.2320

0.0424

                    Table 4. Descriptive comparing statistics

Comparing Scores between Group 1 and

Group 2

 V0-V1

t

df

Sig.

Mean Difference

95% Confidence Interval for Mean

Lower Bound

Upper Bound

FMPS

0.5625

59

0.005

-0.01800

-0.08206

0.04606

FMBS

7.4371

59

0.0001

0.27200

0.19882

0.34518

GI

3.2860

59

0.0017

-0.21700

-0.34914

-0.08486

            Comparing Scores between Group 1 and Group 2 indicates a statistically significant (p<0.001) and a beneficial action of the ozonated olive oil.

  1. Discussion

Discussion: it needs a full revision: it is redundant, vague, and inconclusive in the first half. The second half should better fit in the introduction to explain the state of the art of ozonized oils used in dentistry. Otherwise, the authors could reorganize both the introduction and the discussion in a more flow and transparent way. 

Done

Our study shows that the use of ozonated olive oil in oral hygiene has beneficial effects. By decreasing the values of oral indices in patients who used ozonated olive oil, it is confirmed that it is one of the good products to use in oral hygiene.

Through acceleration of healing, reduction of microbial infection, modulation of the inflammation phase, and enzymatic reactions in oxygen metabolism ozonated olive oil represents an efficient and at the same time cheap alternative method that must be used in the dental field [35]. There are a series of studies in which the basic measurements of oral indices do not show significant differences after the use of ozonated olive oil [38-40]. Different components of olive oil other than fatty acids are responsible for the bactericidal action and the application of ozone oil as an adjuvant is effective[41].

      There is a lack of literature regarding the effect of ozone oil versus olive oil on the clinical parameter of periodontal inflammation. Due to its beneficial biological properties, including antimicrobial and immune-stimulating effects, the use of ozonated olive oil in dentistry has opened up new perspectives in the treatment of dental pathologies for patients of all ages.

Thank you very much for review reports and for the extremely useful observations and suggestions!

Kind regards,

Prof.dr. Liliana Sachelarie

Reviewer 2 Report

In the present study, Ramona Feier and colleagues demonstrate positive effects of ozonated olive oil on gingival infections. The study included 71 patients with gum diseases. The study is well designed and shows interesting differences between the benefits of ozonated olive oil and olive oil in dentistry. However, some methodological and substantive points still need some improvement.

Introduction:

The introduction only vaguely explains the etiology of gingivitis. The different microbial species should be explained and the problem of biofilm formation should be addressed. Periodontitis should also be introduced, as it is explicitly mentioned in the following text.

Furthermore, I miss a detailed explanation of how the problems of oral infections is currently being dealt with. You mention "brushing, flossing, rinsing, and dental hygiene appointments", but this should be specified more.

You also mention that mechanical cleaning can be combined with antibiotic therapy. I totally agree with you, but you should explain this in more detail.

Where the section on conventional therapy is a little too superficial, the introduction to ozone and ozonated oil is a little too redundant. Try to limit yourself to what ozone and ozonated olive oil is, how it works and where it may work in the oral cavity. In lines 91-127 you repeat several times that ozonated olive oil is antibacterial and antimicrobial.

You can go into more detail about the different methods of application in the discussion.

Methods:

In this section you should describe the study design in detail. What is V0, V1? How much time is between visits (3 weeks).

The name, company and country of the available oils should be indicated.

Statistics: Which test did you use?

Results:

You should also show the standard deviation you give in the table as error bars in the graphs.

Statistical differences should also be shown in the figures.

The figure legends should be more detailed.

Discussion:

You describe the possible effects of ozonated oil. Additionally, you present studies by other authors on what ozonated oil can be used for. What is missing, in my opinion, is a proper discussion of your results: You only mention the results of other studies without relating them to the results of your study.

Are there other studies (in vivo or in vitro) that show similar effects of ozonated oil? If so, what was the study design in relation to your study? Are there studies that do not show any positive effects with ozonated oil? What could be the reason for any difference? Is the ozonated olive oil used of a different composition?

Are there other artificial mouth rinses that can be compared to ozonated olive oil?

Author Response

The authors acknowledge the useful observations and suggestions of the reviewer’s as concerns the manuscript entitled:  Comparative study of ozonated olive oil effects on oral hygiene indices

Ramona Feier1, Mircea Radu Sireteanu Cucui1, Ramona Flavia Ratiu1, Dana Baciu1, Carmen Galea1, Liliana Sachelarie 2,*, Claudia Nistor1, Dorin Cocos1, Loredana Liliana Hurjui3, Cernei Eduard Radu3

According to the reviewer’s recommendations, all the suggestions were taken into account, as follows:

In the present study, Ramona Feier and colleagues demonstrate positive effects of ozonated olive oil on gingival infections. The study included 71 patients with gum diseases. The study is well designed and shows interesting differences between the benefits of ozonated olive oil and olive oil in dentistry. However, some methodological and substantive points still need some improvement.

Introduction:

The introduction only vaguely explains the etiology of gingivitis. The different microbial species should be explained and the problem of biofilm formation should be addressed. Periodontitis should also be introduced, as it is explicitly mentioned in the following text.

One of the most common inflammatory diseases of the gingival tissue is gingivitis caused by bacterial infection. This is located at the level of soft gingival tissue and conjunctive tissue. The most common variant encountered in the population is the one caused by plaque.

Clinically this presents as swelling, redness, tenderness, a shiny surface and bleeding on gentle probing. The manifestation is painless so that many patients often do not recognize the disease and fail to seek attention. Potentially associated with gingivitis are species of microorganisms such as Streptococcus, Fusobacterium, Actinomyces, Veillonella and Treponema,  Bacteroides, Capnocytophaga and Eikenella. Of course, there are other etiological factors that favor faster plaque deposition or the tissue's vulnerability to microbial attack. Some of the types of bacteria found within oral infections are part of the microbiota of the oral cavity. As the periodontal disease gingivitis is more common in men than in women, because women have better oral care.

Furthermore, I miss a detailed explanation of how the problems of oral infections is currently being dealt with. You mention "brushing, flossing, rinsing, and dental hygiene appointments", but this should be specified more.

In the treatment of oral infections, the correct choice of antibacterial products has a double benefit, namely the rapid eradication of the infection, which decreases the degree of tissue destruction; as well as decreasing the use of inappropriate antibiotics to prevent the development of antibiotic resistance. Antibiotics containing the narrowest spectrum should be selected based on the results of culture and susceptibility testing.

You also mention that mechanical cleaning can be combined with antibiotic therapy. I totally agree with you, but you should explain this in more detail.

Antibiotic therapy is a powerful adjunct to conventional mechanical debridement for the therapeutic management of periodontal disease. Mechanical treatment and combined surgical treatment can halt or prevent further periodontal attachment loss in most individuals by reducing the total supra-subgingival bacterial mass. Mechanical periodontal treatment supports the host's defense system in overcoming the infection by killing subgingival pathogens, and systemic antibiotics can significantly enhance the effects of mechanical periodontal therapy in combination with measures that improve oral hygiene.

Where the section on conventional therapy is a little too superficial, the introduction to ozone and ozonated oil is a little too redundant. Try to limit yourself to what ozone and ozonated olive oil is, how it works and where it may work in the oral cavity. In lines 91-127 you repeat several times that ozonated olive oil is antibacterial and antimicrobial.You can go into more detail about the different methods of application in the discussion.     

                Olive oil is a vegetable oil obtained by pressing ripe olives. Ozonated olive oil has changes in its chemical composition and new biological properties. Most of the fatty acids in olive oil, most of them, are monounsaturated, predominantly oleic acid (65-85%). With the exception of fatty acids, olive oil is a source of bioactive compounds such as tocopherols or phenolic compounds, with the exception of fatty acids [1]. In the ozonation process, ozone interacts mainly with carbon-carbon double bonds in unsaturated fatty acids giving peroxides, aldehydes and ozonides, which are considered to be responsible for an improvement in the biological activity of the ozonated oil. The amounts and ratio of the compounds depend on several factors such as ozonation time, process temperature and reactor type. According to the literature data, ozonation increases the peroxide value with the amount of ozone introduced to the matrix.

Methods:

In this section you should describe the study design in detail. What is V0, V1? How much time is between visits (3 weeks).

In each patient, we analyzed the daily changes of the index, the Full Mouth Plaque Score (FMPS - the presence of plaque is scored as 1, the absence as 0; the average plaque value is shown as a percentage - % of the sites detected as positive for the presence of plaque. out of a total of 6 sites per tooth), the Full Mouth Bleeding Score (FMBS - the average value of bleeding is indicated as a percentage of the bleeding sites from the total number of sites examined - 6 per tooth) and the gingival index (GI - a value of to 0 to 3 is assigned to the four sites of each element mesial, distal, buccal and lingual/palatal).

  The oral indices were analyzed at the first consultation, marked V0, and after the treatment with ozonated olive oil, three weeks apart, at a new consultation V1.

All patients had significant plaque accumulation as evidenced by the FMPS, despite reporting regular use of a toothbrush 3 times a day and flossing or interdental brushing once a day for interproximal hygiene. FMBS values, the index of inflammation in the active phase, are also above the threshold value, of 20%.

The name, company and country of the available oils should be indicated.

The available products were used in this study: ozonized olive oil (Ozon Relive) and organic cold-pressed extra virgin olive oil from Romania.

Statistics: Which test did you use?

The statistical analysis was performed using IBM SPSS statistics software version 20. Using T Test, descriptive statistics were performed before and after using ozonized olive oil. Data are presented as mean ± standard deviation (SD) or percentage for categorical variables and p-value for statistical significance.

Results:

You should also show the standard deviation you give in the table as error bars in the graphs.

Statistical differences should also be shown in the figures.

Done

Descriptives  statistics t-test

Ozonated olive oil    

Group 1

FMPS

N

Mean

Std. Deviation

SEM

95% Confidence Interval for Mean

Lower Bound

Upper Bound

 t

P value

df

FMPS0

31

112.10700

150.06167

33.55481

22.90983

159.50017

2.7035

0.0102

38

FMPS1

31

20.90200

15.62014

3.49277

FMBS 0

31

40.59100

17.84195

3.25748

20.12495

34.28505

7.6916

0.0001

58

FMBS 1

31

13.38600

7.54811

1.37809

GI 0

31

1.3094

0.3028

0.0535

0.3119

0.6040

6.2758

0.0001

58

GI 1

31

0.8514

0.2560

0.0484

                Statistical analysis of oral index values of group 1 from the first consultation (Vo) to the second consultation after three weeks (V1). P VALUE -this difference is statistically significant. (p<0.001)

Descriptives  statistics t-test

Olive oil    

Group 2

FMPS

N

Mean

Std. Deviation

SEM

95% Confidence Interval for Mean

Lower Bound

Upper Bound

 t

P value

df

FMPS0

3

150.96484

282.59950

50.75637

19.96657

119.9935

2.3988

0.0195

61

FMPS1

30

30.97125

15.65427

2.76731

FMBS 0

30

37.24516

20.18458

3.62526

6.27238

24.69601

3.3623

0.0014

60

FMBS 1

30

21.76097

15.81244

2.84000

GI 0

30

1.3260

0.2796

0.0510

0.1002

0.3658

3.5128

0.0009

58

GI 1

30

1.0930

0.2320

0.0424

Statistical analysis of oral index values of group 2 from the first consultation (Vo) to the second consultation after three weeks (V1).

Comparing Scores between Group 1 and Group 2

V0-V1

t

df

Sig.

Mean Difference

95% Confidence Interval for Mean

Lower Bound

Upper Bound

FMPS

0.5625

59

0.005

-0.01800

-0.08206

0.04606

FMBS

7.4371

59

0.0001

0.27200

0.19882

0.34518

GI

3.2860

59

0.0017

-0.21700

-0.34914

-0.08486

Comparing Scores between Group 1 and Group 2 indicates a statistically significant (p<0.001) and a beneficial action of the ozonated olive oil.

The figure legends should be more detailed.

Done

Discussion:

You describe the possible effects of ozonated oil. Additionally, you present studies by other authors on what ozonated oil can be used for. What is missing, in my opinion, is a proper discussion of your results: You only mention the results of other studies without relating them to the results of your study.

Are there other studies (in vivo or in vitro) that show similar effects of ozonated oil? If so, what was the study design in relation to your study? Are there studies that do not show any positive effects with ozonated oil? What could be the reason for any difference? Is the ozonated olive oil used of a different composition?

Are there other artificial mouth rinses that can be compared to ozonated olive oil?

Our study shows that the use of ozonated olive oil in oral hygiene has beneficial effects. By decreasing the values of oral indices in patients who used ozonated olive oil, it is confirmed that it is one of the good products to use in oral hygiene.

Through acceleration of healing, reduction of microbial infection, modulation of the inflammation phase, and enzymatic reactions in oxygen metabolism ozonated olive oil represents an efficient and at the same time cheap alternative method that must be used in the dental field [35].

There are a series of studies in which the basic measurements of oral indices do not show significant differences after the use of ozonated olive oil [32]. Different components of olive oil other than fatty acids are responsible for the bactericidal action and the application of ozone oil as an adjuvant is effective[36].

      There is a lack of literature regarding the effect of ozone oil versus olive oil on the clinical parameter of periodontal inflammation. Due to its beneficial biological properties, including antimicrobial and immune-stimulating effects, the use of ozonated olive oil in dentistry has opened up new perspectives in the treatment of dental pathologies for patients of all ages.

Thank you very much for review reports and for the extremely useful observations and suggestions!

Kind regards,

Prof.dr. Liliana Sachelarie

Reviewer 3 Report

The title is misleading. The paper is not about "Antibacterial and anti-inflammatory effects of ozonated olive oil in dentistry". The paper compares the effect of mouth rinse with olive oil on clinical signs of gingivitis to ozonated olive oil.

The Antibacterial effect is not assessed in the research. No bacterial count was done. 

The GI count is not clearly described. Reader that wants to replicate your research can not do it. The minimum and maximum value of the index is not mentioned, nor what means 1.3. Is 0.415 important difference? 

The Figures are confusing and not needed. The two tables contain the relevant data. You use comma in some tables instead of decimal point. This is confusing too.

The Discussion should start with the last paragraph. It is dealing with issues outside the research. It should focus on discussing the results, their meaning. what the used scores show and what not, compare the results to other similar experiments.

Author Response

The authors acknowledge the useful observations and suggestions of the reviewer’s as concerns the manuscript entitled:  Comparative study of ozonated olive oil and extra virgin olive oil effects on oral hygiene

Ramona Feier1, Mircea Radu Sireteanu Cucui1, Ramona Flavia Ratiu1, Dana Baciu1, Carmen Galea1, Liliana Sachelarie 2,*, Claudia Nistor1, Dorin Cocos1, Loredana Liliana Hurjui3, Cernei Eduard Radu3

According to the reviewer’s recommendations, all the suggestions were taken into account, as follows:

The title is misleading. The paper is not about "Antibacterial and anti-inflammatory effects of ozonated olive oil in dentistry". The paper compares the effect of mouth rinse with olive oil on clinical signs of gingivitis to ozonated olive oil.

Title

Comparative study of ozonated olive oil and extra virgin olive oil effects on oral hygiene

The Antibacterial effect is not assessed in the research. No bacterial count was done. 

The GI count is not clearly described. Reader that wants to replicate your research can not do it. The minimum and maximum value of the index is not mentioned, nor what means 1.3. Is 0.415 important difference? 

In the case of patients who used the ozonated oil, the mean value of the GI index decreased from 1.291 (at V0) to 0.876 (V1) with an absolute improvement of 0.415. In the case of patients who used the pure oil, the average value of the GI index decreased slightly from 1.326 (at V0) to 1.093 (V1) with an absolute improvement of 0.233.

The Figures are confusing and not needed. The two tables contain the relevant data. You use comma in some tables instead of decimal point. This is confusing too.

Done

The Discussion should start with the last paragraph. It is dealing with issues outside the research. It should focus on discussing the results, their meaning. what the used scores show and what not, compare the results to other similar experiments.

One of the most common inflammatory diseases of the gingival tissue is gingivitis caused by bacterial infection. This is located at the level of soft gingival tissue and conjunctive tissue. The most common variant encountered in the population is the one caused by plaque.

Clinically this presents as swelling, redness, tenderness, a shiny surface and bleeding on gentle probing. The manifestation is painless so that many patients often do not recognize the disease and fail to seek attention [3]. Potentially associated with gingivitis are species of microorganisms such as Streptococcus, Fusobacterium, Actinomyces, Veillonella, Treponema, Bacteroides, Capnocytophaga, and Eikenella. Of course, there are other etiological factors that favor faster plaque deposition or the tissue's vulnerability to microbial attack [3,4].Some of the types of bacteria found within oral infections are part of the microbiota of the oral cavity. As the periodontal disease gingivitis is more common in men than in women, because women have better oral care [3].

      One of the main reasons for the appearance of gingivitis is the accumulation of bacterial plaque. Excess bacterial plaque left due to poor dental hygiene implies the appearance of dental plaque. An incorrect practice of dental hygiene involves the appearance of oral diseases [3]. In the treatment of oral infections, the correct choice of antibacterial products has a double benefit, namely the rapid eradication of the infection, which decreases the degree of tissue destruction; as well as decreasing the use of inappropriate antibiotics to prevent the development of antibiotic resistance. Antibiotics containing the narrowest spectrum should be selected based on the results of culture and susceptibility testing [5,6].

Ozonated olive oil has changes in its chemical composition and new biological properties. Most of the fatty acids in olive oil, most of them, are monounsaturated, predominantly oleic acid (65-85%). With the exception of fatty acids, olive oil is a source of bioactive compounds such as tocopherols or phenolic compounds, with the exception of fatty acids [2]. In the ozonation process, ozone interacts mainly with carbon-carbon double bonds in unsaturated fatty acids giving peroxides, aldehydes and ozonides, which are considered to be responsible for an improvement in the biological activity of the ozonated oil. The amounts and ratio of the compounds depend on several factors such as ozonation time, process temperature and reactor type. According to the literature data, ozonation increases the peroxide value with the amount of ozone introduced to the matrix [10,11].

 To evidence the effect of the therapy, the differences of the variables at time V1 and at time V0 are taken into account ( Table 2, Table 3, Table 4).

                    Table 2. Descriptive statistics for group 1

Descriptives  statistics t-test

Ozonated olive oil     Group 1

N

Mean

Std. Deviation

SEM

95% Confidence Interval for Mean

Lower Bound

Upper Bound

 t

P value

df

FMPS 0

31

112.10700

150.06167

33.55481

22.90983

159.50017

2.7035

0.0102

58

FMPS 1

31

20.90200

15.62014

3.49277

FMBS 0

31

40.59100

17.84195

3.25748

20.12495

34.28505

7.6916

0.0001

58

FMBS 1

31

13.38600

7.54811

1.37809

GI 0

31

1.3094

0.3028

0.0535

0.3119

0.6040

6.2758

0.0001

58

GI 1

31

0.8514

0.2560

0.0484

                                                                                Statistical analysis of oral index values of group 1 from the first consultation (Vo) to the                                                                         second consultation after three weeks (V1) indicates a very statistically significant                                                                                                 (p<0.001).

Table 3. Descriptive statistics for group 2

Descriptives  statistics t-test

Olive oil    

Group 2

N

Mean

Std. Deviation

SEM

95% Confidence Interval for Mean

Lower Bound

Upper Bound

 t

P value

df

FMPS0

30

150.96484

282.59950

50.75637

19.96657

119.9935

2.3988

0.0195

61

FMPS1

30

30.97125

15.65427

2.76731

FMBS 0

30

37.24516

20.18458

3.62526

6.27238

24.69601

3.3623

0.0014

60

FMBS 1

30

21.76097

15.81244

2.84000

GI 0

30

1.3260

0.2796

0.0510

0.1002

0.3658

3.5128

0.0009

58

GI 1

30

1.0930

0.2320

0.0424

                    Table 4. Descriptive comparing statistics

Comparing Scores between Group 1 and

Group 2

 V0-V1

t

df

Sig.

Mean Difference

95% Confidence Interval for Mean

Lower Bound

Upper Bound

FMPS

0.5625

59

0.005

-0.01800

-0.08206

0.04606

FMBS

7.4371

59

0.0001

0.27200

0.19882

0.34518

GI

3.2860

59

0.0017

-0.21700

-0.34914

-0.08486

      Comparing Scores between Group 1 and Group 2 indicates a statistically significant (p<0.001) and a beneficial action of the ozonated olive oil.

The Discussion should start with the last paragraph. It is dealing with issues outside the research. It should focus on discussing the results, their meaning. what the used scores show and what not, compare the results to other similar experiments.

Through acceleration of healing, reduction of microbial infection, modulation of the inflammation phase, and enzymatic reactions in oxygen metabolism ozonated olive oil represents an efficient and at the same time cheap alternative method that must be used in the dental field [35]. There are a series of studies in which the basic measurements of oral indices do not show significant differences after the use of ozonated olive oil [38-40]. Different components of olive oil other than fatty acids are responsible for the bactericidal action and the application of ozone oil as an adjuvant is effective[41].

      There is a lack of literature regarding the effect of ozone oil versus olive oil on the clinical parameter of periodontal inflammation. Due to its beneficial biological properties, including antimicrobial and immune-stimulating effects, the use of ozonated olive oil in dentistry has opened up new perspectives in the treatment of dental pathologies for patients of all ages.

Thank you very much for review reports and for the extremely useful observations and suggestions!

Kind regards,

Prof.dr.Liliana Sachelarie

Round 2

Reviewer 1 Report

The paper has improved significantly. However, some caveats remain unresolved and require further review process:

1.     Moderate English changes are required: tenses and syntax need to be corrected along with the manuscript. I suggest authors refer to a native English speaker or specific tools, such as Grammarly, which is available in a version free from cost, otherwise a professional editorial assistant.

2.     As expected, the work can not be improved by adding two other positive and negative control groups. This issue must be stated in the discussion as a study limitation.

3.     In the discussion section, the authors should also better discuss the state of the art and previous literature bout the efficacy of ozonized olive oils in dentistry and periodontology. For this purpose, but not mandatory, the following paper from which to draw inspiration:

·      PMID: 36110593 

·      PMID: 33050423

·      PMID: 32932898 

·      PMID: 30607228

·      PMID: 30369807

4.     The authors did not address all the required revisions, such s the following unsolved points:

In introduction:

-        The limitations of the common substances (chlorhexidine and other antiseptics)

-        The literature report (state of the art) on ozonated oil in dentistry: the authors reported only experiences with ozone therapy (ref. 12-15, 18, 20-28). For this purpose, see the suggestion above. This content can be developed in the introduction or the discussion at your convenience.

5.     The authors reported, “The aim of this study is to evaluate the efficacy of ozonated olive oil in dental hygiene by comparing it with pure olive oil.”. It should be discussed if the promising results from their experiments should be related to the “ozone” rather than oil. Or why ozonated oil could be better than ozone alone. For these reasons, a control group treated with other compounds or placebo may have been helpful. However, at this point, the authors should argue in the “discussion section” if the promising results from their experiments should be related to the “ozone” rather than ozonated oil.

6.     In the discussion, the authors added, There are a series of studies in which the basic measurements of oral indices do not show significant differences after the use of ozonated olive oil [38-40]. ”. So, the authors should explain why their results differ from the cited study. Is the ozonized oil useful or not?

Author Response

The authors acknowledge the useful observations and suggestions of the reviewer’s as concerns the manuscript entitled:  Comparative study of ozonated olive oil effects in oral hygiene

Ramona Feier1, Mircea Radu Sireteanu Cucui1, Ramona Flavia Ratiu1, Dana Baciu1, Carmen Galea1, Liliana Sachelarie 2,*, Claudia Nistor1, Dorin Cocos1, Loredana Liliana Hurjui3, Cernei Eduard Radu3

According to the reviewer’s recommendations, all the suggestions were taken into account, as follows:

  1. Moderate English changes are required: tenses and syntax need to be corrected along with the manuscript. I suggest authors refer to a native English speaker or specific tools, such as Grammarly, which is available in a version free from cost, otherwise a professional editorial assistant.

                        Done

  1. As expected, the work can not be improved by adding two other positive and negative control groups. This issue must be stated in the discussion as a study limitation.

One of the study's limitations is the small number of groups of study.

  1. In the discussion section, the authors should also better discuss the state of the art and previous literature bout the efficacy of ozonized olive oils in dentistry and periodontology. For this purpose, but not mandatory, the following paper from which to draw inspiration:
  • PMID: 36110593 
  • PMID: 33050423
  • PMID: 32932898 
  • PMID: 30607228
  • PMID: 30369807

     The therapeutic action of ozonated olive oil is its effectiveness in reducing periodontal indices. Subjects using ozonated olive oil for teeth have noted good responses in diminishing the rate of tooth decay, gingivitis, and chapped lips and ozonated olive oil could be considered an alternative antibacterial agent [28,29].

There are also studies that have shown that ozonated olive oil promotes the healing of grafts with a very good prognosis [30]. A particularly important aspect in dentistry is the prevention of various diseases. There are studies that indicate that ozonated olive oil has the effect of improving the results of scaling and root planing (SRP) in the treatment of periodontal diseases [31,32]. Compliance with tooth brushing procedures and the use of mouthwash with ozonated olive oil could be an additional preventive tool for the oral hygiene procedure at home. Further research is needed to evaluate the effectiveness of ozonated olive oil mouthwashes in patients at high risk of caries and in subjects requiring orthodontic therapy [33-35]. A particularly important aspect in dentistry is the prevention of various diseases. Compliance with tooth brushing procedures and the use of mouthwash with ozonated olive oil could be an additional preventive tool for the oral hygiene procedure at home.

      One of the study's limitations is the small number of groups of study. Further research is needed to evaluate the effectiveness of ozonated olive oil mouthwashes in patients at high risk of caries and in subjects requiring orthodontic therapy [36,37].

  1. The authors did not address all the required revisions, such s the following unsolved points:

In introduction:

-        The limitations of the common substances (chlorhexidine and other antiseptics)

                                Preventing and stopping microbial development in the oral cavity can be                   achieved by using antimicrobial solutions like Chlorhexidine (CHX)  a                                                commonly used antiseptic mouthwash used by dentists and the public                                           for its antimicrobial effects [9].

-        The literature report (state of the art) on ozonated oil in dentistry: the authors reported only experiences with ozone therapy (ref. 12-15, 18, 20-28). For this purpose, see the suggestion above. This content can be developed in the introduction or the discussion at your convenience.

Done

  1. The authors reported, “The aim of this study is to evaluate the efficacy of ozonated olive oil in dental hygiene by comparing it with pure olive oil.”. It should be discussed if the promising results from their experiments should be related to the “ozone” rather than oil. Or why ozonated oil could be better than ozone alone. For these reasons, a control group treated with other compounds or placebo may have been helpful. However, at this point, the authors should argue in the “discussion section” if the promising results from their experiments should be related to the “ozone” rather than ozonated oil.

      Through acceleration of healing, reduction of microbial infection, modulation of the inflammation phase, and enzymatic reactions in oxygen metabolism ozonated olive oil represents an efficient and at the same time cheap alternative method that must be used in the dental field [38-40].

      The results obtained are given by the different components of olive oil, responsible for the bactericidal action and for skin absorption because it can stabilize ozone, which is a very reactive molecule [34]. Ozonated olive oil combines the properties of ozone with those of olive oil to achieve an unparalleled compound. The oxidizing power of ozone has interesting effects on microorganisms and wound healing [41-43].

  1. In the discussion, the authors added, “There are a series of studies in which the basic measurements of oral indices do not show significant differences after the use of ozonated olive oil [38-40]. ” So, the authors should explain why their results differ from the cited study. Is the ozonized oil useful or not?

This phrase does not appear in the revised document. Thank you!

Due to its beneficial biological properties, including antimicrobial and immune-stimulating effects, the use of ozonated olive oil in dentistry has opened up new perspectives in the treatment of dental pathologies for patients of all ages.

Thank you very much for your review,

Respectfully,  

Prof. dr. Liliana Sachelarie

Reviewer 2 Report

Thank you for the revision of the article, which has improved the readability. Even though the topic is very interesting and there are not many comparable studies, in my opinion, the article should be further revised before it can be published in Applied Sciences.

1)      I still miss a clear definition of periodontits in the introduction

2)      You write: “As the periodontal disease gingivitis is more common in men than in women, because women have better oral care.” This is not supported by the reference, you cite (3)

3)      The authors still don’t show the Name / Company / country of the products, used in the study. It must be presented like: olive oil (Ozonrelieve, Rome, Italy)

4)      Overall, the figure legends are not yet described in sufficient detail. The legends should explain the figures in such a way that they can be understood without the text. I also do not understand Table 4 from this point of view.

5)      You explain possible antimicrobial mechanisms of oil and ozonated oil in the discussion. But again, I miss a real discussion about your results. You write, that “there are a series of studies in which the basic measurements of oral indices do not show significant differences after the use of ozonated olive oil”. This should be discussed with regard to your results. Are there differences in study population, observation times, data collection, materials, etc...? What might be the reason?

Author Response

The authors acknowledge the useful observations and suggestions of the reviewer’s as concerns the manuscript entitled:  Comparative study of ozonated olive oil effects in oral hygiene

Ramona Feier1, Mircea Radu Sireteanu Cucui1, Ramona Flavia Ratiu1, Dana Baciu1, Carmen Galea1, Liliana Sachelarie 2,*, Claudia Nistor1, Dorin Cocos1, Loredana Liliana Hurjui3, Cernei Eduard Radu3

According to the reviewer’s recommendations, all the suggestions were taken into account, as follows:

Thank you for the revision of the article, which has improved the readability. Even though the topic is very interesting and there are not many comparable studies, in my opinion, the article should be further revised before it can be published in Applied Sciences.

  • I still miss a clear definition of periodontits in the introduction

Periodontitis is a serious infection of the gums that damages the soft tissue and, without treatment, can cause tooth loosening or lead to tooth loss [4].

  • You write: “As the periodontal disease gingivitis is more common in men than in women, because women have better oral care.” This is not supported by the reference, you cite (3)

As the periodontal disease gingivitis is more common in men than in women, because women have better oral care [12].

  • The authors still don’t show the Name / Company / country of the products, used in the study. It must be presented like: olive oil (Ozonrelieve, Rome, Italy)

The available products were used in this study: ozonized olive oil (Ozonrelieve, Rome, Italy) and organic cold-pressed extra virgin olive oil.

  • Overall, the figure legends are not yet described in sufficient detail. The legends should explain the figures in such a way that they can be understood without the text. I also do not understand Table 4 from this point of view.

Done

  • You explain possible antimicrobial mechanisms of oil and ozonated oil in the discussion. But again, I miss a real discussion about your results. You write, that “there are a series of studies in which the basic measurements of oral indices do not show significant differences after the use of ozonated olive oil”. This should be discussed with regard to your results. Are there differences in study population, observation times, data collection, materials, etc...? What might be the reason?

This phrase does not appear in the revised document. Thank you very much!

Due to its beneficial biological properties, including antimicrobial and immune-stimulating effects, the use of ozonated olive oil in dentistry has opened up new perspectives in the treatment of dental pathologies for patients of all ages.

Thank you very much for your review,

Respectfully,

Prof.dr. Liliana Sachelarie
